# Genomics-informed isolation and characterization of a symbiotic Nanoarchaeota system from a terrestrial geothermal environment

Louie Wurch[1,2,†], Richard J. Giannone[1], Bernard S. Belisle[1,2], Carolyn Swift[1,2], Sagar Utturkar[1], Robert L. Hettich[1,2], Anna-Louise Reysenbach[3] & Mircea Podar[1,2]

Biological features can be inferred, based on genomic data, for many microbial lineages that remain uncultured. However, cultivation is important for characterizing an organism's physiology and testing its genome-encoded potential. Here we use single-cell genomics to infer cultivation conditions for the isolation of an ectosymbiotic Nanoarchaeota ('*Nanopusillus acidilobi*') and its host (*Acidilobus*, a crenarchaeote) from a terrestrial geothermal environment. The cells of '*Nanopusillus*' are among the smallest known cellular organisms (100–300 nm). They appear to have a complete genetic information processing machinery, but lack almost all primary biosynthetic functions as well as respiration and ATP synthesis. Genomic and proteomic comparison with its distant relative, the marine *Nanoarchaeum equitans* illustrate an ancient, common evolutionary history of adaptation of the Nanoarchaeota to ectosymbiosis, so far unique among the Archaea.

[1] Oak Ridge National Laboratory, Oak Ridge, Tennessee 37831, USA. [2] Department of Microbiology, University of Tennessee, Knoxville, Tennessee 37996, USA. [3] Department of Biology, Portland State University, Portland, Oregon 97207, USA. † Present address: Department of Biology, James Madison University, Harrisonburg, Virginia 22807, USA. Correspondence and requests for materials should be addressed to M.P. (email: podarm@ornl.gov).

Culture-independent approaches have revolutionized our understanding of microbial diversity and evolution. Metagenomics and single-cell genomics enable inferences of metabolic traits and involvement in specific biogeochemical transformations by uncultured microbes that remain elusive to direct investigation[1–4]. In some cases, physical identification of such microbes in environmental samples have led to characterization of some of their structural and molecular features that better explain important ecological properties, such as biofilm formation or syntrophic associations[5–7]. Nevertheless, laboratory cultures remain essential for detailed investigations of organismal biology, including mechanisms of interspecies interactions. So far, however, there have been only a couple examples of genome-informed isolation of novel microbes, in which inferred phenotypic traits were used to select effective cultivation conditions[8,9].

Nanoarchaeota has been a proposed archaeal lineage following the discovery and cultivation of *Nanoarchaeum equitans*, an ultra-small ectosymbiont residing on the marine hyperthermophilic crenarchaeote, *Ignicoccus hospitalis*[10]. Ribosomal RNA gene surveys revealed that related nanoarchaea are widespread in marine and terrestrial thermal environments around the world but none have been isolated[11–16]. Genomic and physiological features of *N. equitans* (extensive fragmentation of protein and tRNA genes, massive loss of biosynthetic capabilities) suggest a parasitic-type adaptation that explains its strict dependence on direct contact with its host[17–21]. Using a single-cell genomics approach we recently sequenced the near-complete genome of a second, yet uncultured member of the Nanoarchaeota (Nst1) together with its host, an uncultured Sulfolobales crenarchaeote from Obsidian Pool in Yellowstone National Park (YNP)[22]. Similar to *N. equitans*, Nst1 displays extensive genome reduction and lacks most primary biosynthetic and energetic functionalities, but has retained potential glycolysis and/or gluconeogenesis pathways.

The origins of Nanoarchaeota remain unclear as their reduced gene repertoire and rapidly evolving protein sequences preclude a firm phylogenetic placement in the archaeal tree. Recent phylogenetic analyses suggest relationships with other distinct lineages of ultra-small, exclusively uncultured Archaea, sometimes generically coined 'nanoarchaea' (Parvarchaea, Nanohaloarchaea, Aenigmarchaeota and Diapherotrites), all with ties to the Euryarchaeaota[2,23,24]. A better understanding of how Nanoarchaeota evolved to be dependent on direct interaction with other Archaea and how that relationship shaped the genomes and physiology of the interacting species clearly requires the isolation and detailed characterization of such novel symbiotic systems. Because the interacting organisms in these systems have small, similarly sized genomes and physiological complexity, they can also serve as models to understand molecular and evolutionary mechanisms of symbiosis in single-celled organisms.

Here we use single-cell genomics to infer physiological dependencies and adaptation to low-pH environments for the isolation and characterization of the first terrestrial Nanoarchaeota-host system, from an acidic hot spring in YNP.

## Results

**Isolation and characterization of a geothermal Nanoarchaeota system.** Using small subunit rRNA gene amplification and 454 pyrosequencing, we detected the relatively abundant presence of Nanoarchaeota (7% of total archaeal sequences) in Cistern Spring, a mildly acidic thermal spring (pH 4.5, 82 °C) from the Norris Geyser Basin of YNP (Methods). On the basis of the previous findings that Nanoarchaeota in such environments likely use extreme thermoacidophilic Crenarchaeota as hosts[22,25] and that they may be able to utilize peptides or complex sugars[22], we established enrichment cultures that would select for their growth. Stable presence of Nanoarchaeota was achieved in anaerobic Brock medium[26] supplemented with yeast extract, casamino acids, sucrose or glycogen at temperatures of 80–85 °C and pH 3–3.5. Following continued increase in abundance after serial passaging, as determined by a quantitative PCR (qPCR) assay, Nanoarchaeota cells were microscopically observed by 4,6-diamidino-2-phenylindole (DAPI) staining as tiny cells on the surface of larger, ∼1 μm discoid cells. Taxonomic characterization of the enriched community revealed the abundant presence of *Acidilobus*, a Desulfurococcales crenarchaeote. The association of the Nanoarchaeota with *Acidilobus* was confirmed using single-cell sorting followed by multiple displacement amplification and sequencing of 16S rRNA-amplified genes. Following two rounds of dilution to extinction, optical tweezer selection and inoculation of a single discoid cell carrying a nanoarchaeon into liquid medium, we obtained a pure co-culture of *Acidilobus* sp. 7A and its nanoarchaeon, strain N7A. Fluorescence *in situ* hybridization revealed that these tiny Nanoarchaeota are present in variable numbers on *Acidilobus* host cells (Fig. 1) and, occasionally, as free cells.

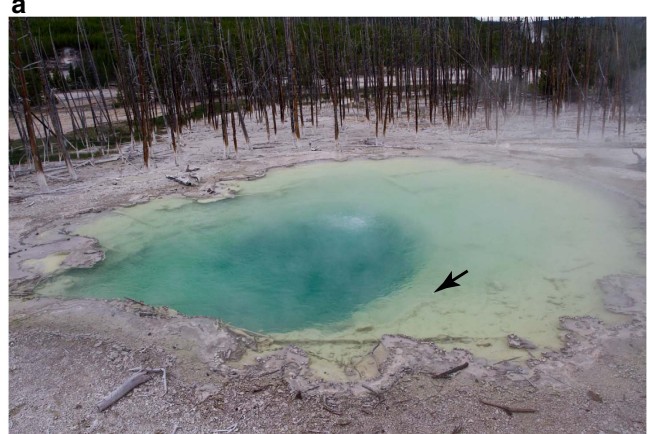

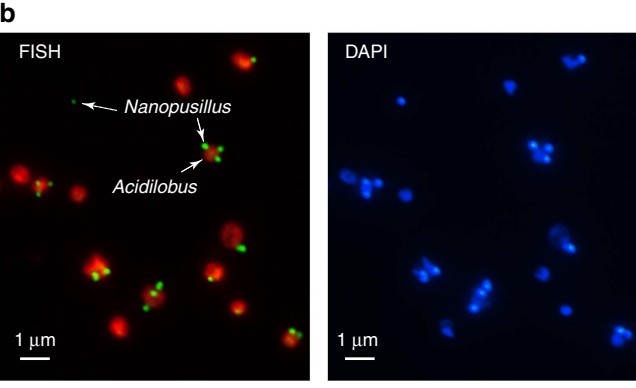

**Figure 1 | Environment sampling for Nanoarchaeota cultivation.** (**a**) Cistern Spring pool in YNP. The arrow points to the sampled site for the Nanoarchaeota enrichments. (**b**) Fluorescence *in situ* hybridization (FISH) of a *N. acidilobi*–*Acidilobus* sp. 7A co-culture using Nanoarchaeota (green) and Crenarchaeota (orange) probes. The dotted arrow points to a free Nanoarchaeota cell. On the right is the same field imaged for DNA staining with DAPI.

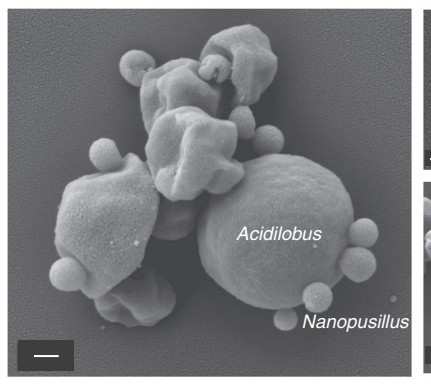
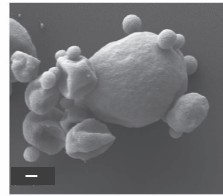

**Figure 2 | Scanning electron micrographs of *N. acidilobi* on *Acidilobus* sp. 7A cells in co-culture.** The arrow points to the place of contact between two cells that reveals membrane stretching. Scale bars, 200 nm.

On the basis of 16S rRNA gene sequence, strain N7A is closely related (97–98% identity) to uncultured Nanoarchaeota from several Yellowstone geothermal environments[12,15,25], including Nst1 from Obsidian Pool[22]. Comparatively, the marine species *N. equitans* shares 80% nucleotide identity in the 16S rRNA gene, suggesting that the marine and terrestrial lineages belong to distinct orders of the Nanoarchaeota. Scanning electron micrographs revealed that the N7A nanoarchaeon cells are smaller (100–300 nm in diameter) than *N. equitans* (300–400 nm) (Fig. 2), similar to ultra-small uncultured bacteria observed in subsurface environments[27], and may be the smallest cultured organism to date. As strain N7A is the first isolated member of geothermal Nanoarchaeota, representing a new genus and species and probably new order within the Nanoarchaeota, we propose the following Candidatus status for this taxon.

'***Nanopusillus acidilobi***' gen. et sp. nov.

**Etymology**. *Nanus* (Latin adj.) 'dwarf', referring to its size and denote placing within the Nanoarchaeota; *pu'sil.lus* (Latin adj.) very small, indicating its extremely small size, at the boundary of cellular life; *a.ci.di.lo'bi* (Latin n. gen.) of acidilobus, growth dependent on Acidilobus.

**Locality**. Cistern Spring pool (water and sediment slurry), in the Norris Geyser basin of YNP (latitude: 44.723; longitude: −110.704″).

**Diagnosis**. coccoid cells, 100–300 nm in diameter, obligate ectosymbionts/parasites on the surface of the thermoacidophilic crenarchaeote Acidilobus. Occasional free cells can be observed in the co-culture but their viability is unknown. Optimum growth is in co-culture with its host at 82 °C and pH 3.6. First isolated from Cistern Spring, a hot acidic spring in YNP. On the basis of single-cell genomics and metagenomic data, related strains or species that may use other Crenarchaeota as hosts are present in other acidic hot springs in YNP, at pH 2–6 (refs 22,25).

*Acidilobus* sp. 7A, the host of *N. acidilobi,* is nearly identical (99.9%) at the 16S rRNA level to *Acidilobus sulfurireducens* 18D70, isolated from Dragon Spring in YNP[28], but distinct (95–97%) from other *Acidilobus* species isolated from Kamchatka, Russia[29,30] or identified in metagenomes from several other YNP thermal features, including Cistern Spring[31](Fig. 3). Similar to *A. sulfurireducens*, *Acidilobus* 7A is a thermoacidophilic anaerobe, with an optimal growth temperature of 80–82 °C at pH 3.5 (Supplementary Fig. 1), that relies on peptides or polysaccharides as carbon source. *A. sulfurireducens* was reported to be strictly dependent on sulfur, possibly as soluble nano-sized particles, which acts as a terminal electron acceptor[28,32]. In contrast, *Acidilobus* 7A is only weakly stimulated by the inclusion of elemental sulfur or thiosulfate in the media (Supplementary Fig. 2), being able to derive its energy from fermenting organic substrates. Importantly, variations in media composition or other growth conditions did not significantly affect the abundance of *N. acidilobi* in co-culture with its host. Therefore, subsequent cultivation and characterization of this system were performed in anaerobic *Acidilobus* mineral media[28] supplemented with vitamins, yeast extract, peptone and thiosulfate, at pH 3.5 and 82 °C. Because of the extremely small size of the nanoarchaeum and the irregular shape of their host cells, monitoring and counting these cells using direct microscopy was challenging. We therefore developed and employed species-specific qPCR and immunofluorescence microscopy to determine cell densities and relative abundance of *Nanopusillus* on host cells.

In pure culture, *Acidilobus* 7A enters stationary phase after 4–5 days, reaching a density of $\sim 5 \times 10^7$ ml$^{-1}$. This growth profile is not impacted by the presence of the nanoarchaeon in a co-culture (Fig. 4), unlike *I. hospitalis*, whose cell division is inhibited by *N. equitans* cells on its surface[19]. *N. equitans* rapidly divides even after its host reaches stationary phase and is present at densities of 10 cells or more on most *Ignicoccus* cells. In comparison, *N. acidilobi* follows the same growth kinetics with that of its host and in the stationary phase co-culture as many as half the *Acidilobus* cells lack attached ectosymbionts, while others have anywhere from 1–10 *Nanopusillus* cells on their surface. Scanning electron microscopy also revealed the spatial distribution of *N. acidilobi* cells on its host, as well as what appears to be membrane stretching at the point of contact between the two organisms, suggesting strong intercellular interaction (Fig. 2). Nevertheless, we also observed free nanoarchaeal cells in the media, similar to what is the case with *Nanoarchaeum*[19]. It is not known if such detached cells are viable and able to attach to a new host.

In the case of *N. equitans*, it has been shown that Archaea other than *Ignicoccus* cannot serve as the host[19], suggesting that co-evolution has led to specific recognition and interaction mechanisms. To test if *N. acidilobi* is also strictly adapted to its host, we introduced a related *Acidilobus* species, *Acidilobus saccharovorans* in the co-culture, followed by cell staining with antibodies specific for *N. acidilobi* and *Acidilobus* 7A (Fig. 5). We could not detect any attachment of *N. acidilobi* cells to *A. saccharovorans*, suggesting that cross-species host switching does not readily occur in the laboratory. Such a process, however, cannot be ruled out over long co-occurrence in the environment and may explain the relatively low divergence levels across Nanoarchaeota that use different archaeal hosts in YNP[22,25].

**Genomics and proteomics of the *Nanopusillus–Acidilobus* system.** Genomic draft assemblies of *N. acidilobi* and *Acidilobus* 7A were obtained by sequencing both a co-culture of the two organisms and a pure *Acidilobus* 7A culture using the Illumina MiSeq platform. Sequencing data identified a significant difference in the genome GC% between the two organisms (24% for *N. acidilobi* and 58% for *Acidilobus* 7A), which is similar to values reported for other nanoarchaeotal and *Acidilobus* genomes[21,22,31,33]. To purify *N. acidilobi* DNA from its host, we applied ultracentrifugation in a CsCl gradient in the presence of the bisbenzimide, which preferentially binds AT-rich DNA (Supplementary Fig. 3), followed by PacBio long-read sequencing. Hybrid assembly of the MiSeq and PacBio data resulted in closed, circular genomes for both organisms. We also performed a whole-cell, liquid chromatography-tandem mass spectrometry (LC-MS/MS)-based proteomic analysis of a co-culture of *N. acidilobi–Acidilobus* 7A. The proteomics data enabled a more detailed glimpse into which inferred metabolic features are expressed and, based on peptide-spectrum matches and normalized ion intensity values for each protein, their relative abundance.

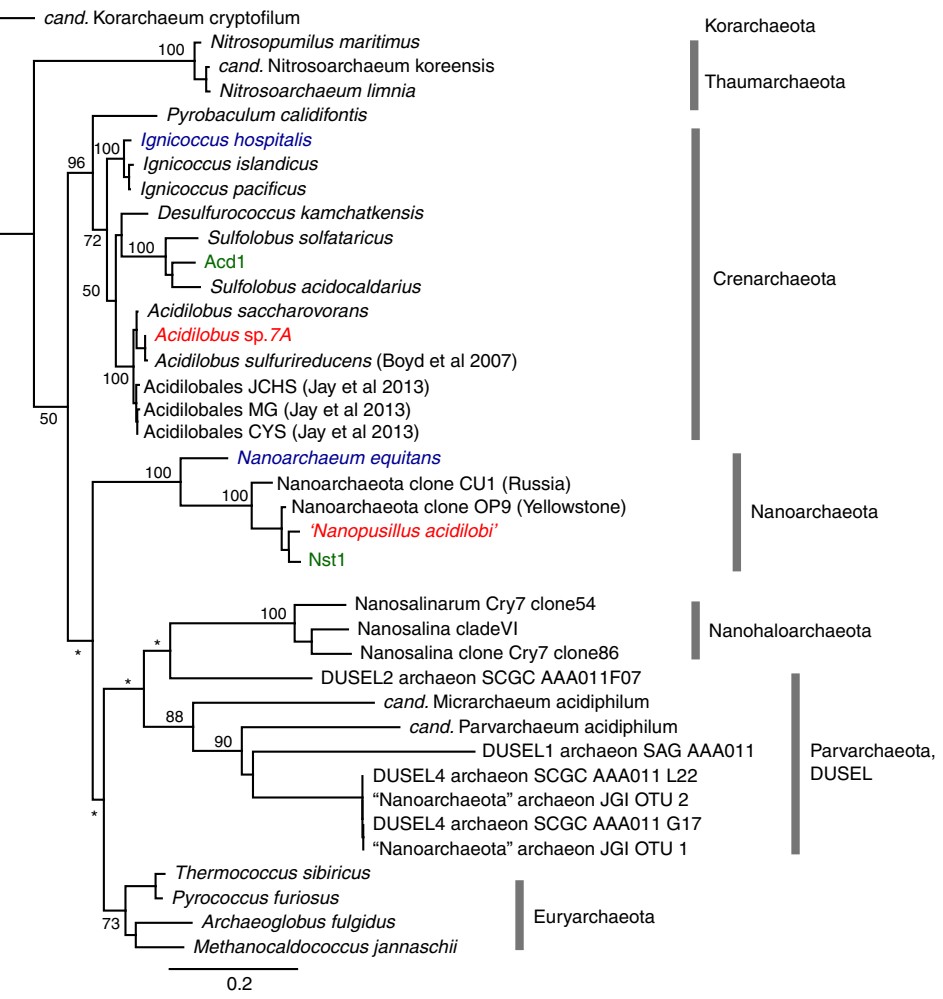

**Figure 3 | Maximum likelihood phylogeny of selected Archaea based on small subunit (SSU) rRNA.** Bootstrap support is indicated for selected groups at the nodes (* if <50). The three Nanoarchaeota-host systems are highlighted using blue, green and red colour.

The genome of *N. acidilobi* is 605,887 nucleotides in length and encodes 656 proteins and 46 RNAs, including 43 tRNAs representing all amino acids. There is extensive synteny with the contigs of the Nst1 single-cell draft genome (Fig. 6). The main genomic features are also similar to those described for Nst1 and its close relative from Yellowstone's Nymph Lake thermal area[22,25]. These include the absence of split tRNA genes that require assembly in *trans*, as has been demonstrated in *N. equitans*[34], and the presence of a common core of split protein genes (Supplementary Table 1). *N. acidilobi* is also incapable of synthesizing most necessary metabolic precursors, including nucleotides, co-factors, amino acids and lipids, suggesting that like *N. equitans*, it must obtain these from its host. Unlike *N. equitans*, however, the genomes of *N. acidilobi* and its relatives encode a surprisingly rich set of enzymes involved in carbohydrate metabolism (discussed in detail in ref. 22). Not surprisingly, considering the relatively large phylogenetic distance, the level of syntheny and sequence similarity to *N. equitans* is modest (Fig. 6).

LC-MS/MS proteomics detected roughly one-third of all predicted *Nanopusillus* proteins and measured their relative abundances, allowing for initial inferences into the different metabolic potentials of these co-interacting species (Fig. 7 and Supplementary Data 1). Two of the diagnostic enzymes for gluconeogenesis, phosphoenolpyruvate synthetase (Nps2680) and the bifunctional fructose-1,6-bisphosphate aldolase/phosphatase (Nps275) were amongst the most highly abundant proteins in the cell, suggesting this process is quite active in *N. acidilobi*. Most of the other enzymes that are involved in archaeal gluconeogenesis[35] were detected as well, except for the glyceraldehyde-3-phosphate dehydrogenase/phosphoglycerate kinase couple, which was not identified in the genome. A potentially reversible ferredoxin-dependent aldehyde oxidoreductase (Nps740) was identified, which could serve as a substitute and catalyse the anabolic reaction. A non-phosphorylating glyceraldehyde-3-phosphate dehydrogenase (Nps2395) would operate in the glycolytic direction. The predicted pyruvate kinase (Nps1700) was not detected in the proteome data, which questions the operability of the glycolytic cycle under the tested culture conditions. Alternatively, phosphoenolpyruvate synthetase, which has been shown in other Archaea to function in glycolysis as well[36,37], may substitute. The source of acetyl-CoA that feeds into the gluconeogenesis pathway remains unknown. Though a potential acetyl-CoA synthase (Nps_2631) is encoded by the genome, its presence was not detected by LC-MS/MS nor does the addition of acetate to the medium affect growth. An alternative hypothesis may be that acetyl-CoA is acquired from the host.

It remains unknown if *N. acidilobi* can perform glycolysis, which would be a source for ATP. If intact, stored glycogen could provide limited energetic independence for cells that would dissociate from their host. The confirmed expression of archaellum (archaeal flagellum) proteins suggests that *N. acidilobi* cells may be capable of motility when detached. The reversed ribulose monophosphate pathway is absent, so glycogen does not appear

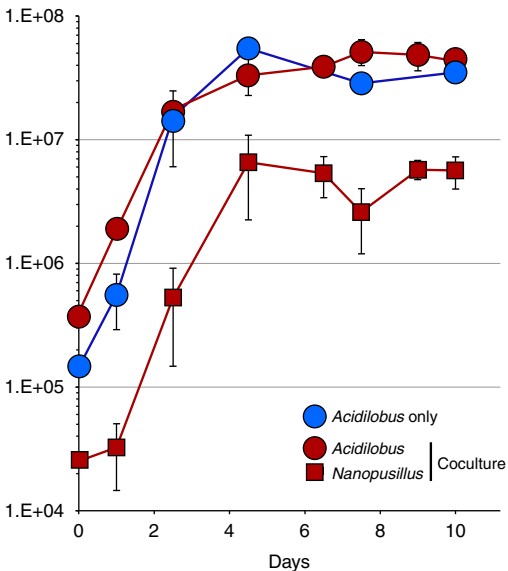

**Figure 4 | Time course growth of *Acidilobus* sp. 7A in pure culture and in co-culture with *Nanopusillus acidilobi*.** Error bars (s.d.) are based on three culture replicates.

to serve for pentose biosynthesis. A likely role for gluconeogenesis appears to be the synthesis of activated sugars that can be used for protein and lipid glycosylation. Enzymes involved in glycogen hydrolysis and several glycosyl transferases were detected in the proteome. To further test if surface glycosylation occurs, we incubated cells in the presence of fluorescently labelled lectins. *N. acidilobi* cells were labelled by wheat germ agglutinin, indicating the surface presence of *N*-acetylglucosamine (Fig. 5). Importantly, we observed differential staining for the host, which was labelled by concanavalin A (α-manose and α-glucose specific) and by the *Dolichos biflorus* lectin (DBA, *N*-acetyl galactosamine specific), which suggests that the activated sugars are not exchanged between the cells. The role of such post-translational modifications in *N. acidilobi* is unknown, but cell surface glycosylation is complex and widespread in Archaea[38]. Considering that proteins involved in carbohydrate metabolism represented nearly 10% of the proteome but do not appear to have energetic function, surface glycosylation in *N. acidilobi* may play an important role, potentially in host recognition and interaction.

Aside from the enzymes involved in gluconeogenesis, which are absent in *N. equitans*, a comparison between the two proteomes revealed similarity amongst the abundant proteins, including chromosomal assembly proteins, chaperones, replication, transcription and translation factors (Supplementary Fig. 4 and Supplementary Data 2)[17]. One example is TrmBL2, a protein that functions both in chromosome assembly and general transcription repression[39], highly abundant both in *N. equitans* (Neq98) and *N. acidilobi* (Nps2735). This is perhaps not totally unexpected since both Nanoarchaeota are subjected to tight control of cell division and cellular processes that enable contact with and metabolite uptake from their host cell. One notable absence from the *N. acidilobi* proteome was the predicted surface S-layer protein (Nps760). Sequence inspection revealed, however, that the low abundance of proteolytic sites and predicted large-size tryptic peptide size distribution renders this protein difficult if not impossible to identify by trypsin-based shotgun proteomics.

The most surprising feature of *N. acidilobi* is the apparent absence of membrane ATP synthase complex, as well as any components of the respiratory chain. The genome of *N. equitans* encodes an incomplete ATP synthase, which, even though shown

to be expressed, has not been demonstrated to either generate or hydrolyze ATP[17,40]. Structural analysis of the core complex (A–B subunits) further suggests the *N. equitans* ATP synthase is inactive[41]. While the ATP synthase is absent in some insect and plant endosymbiotic or parasitic bacteria with extremely reduced genomes[42–44], these organisms retain respiratory complexes to provide energy and maintain a polarized membrane. In addition, these organisms are completely engulfed by the host cytoplasm, offering a large surface area for diverse transport mechanisms between both host and symbiont. This is not the case in Nanoarchaeota, which are extracellular symbionts covered by an S layer that comes into contact with their host over a very small area (40–50 nm in diameter)[20]. Both nanoarchaea encode a minute repertoire of transporters (represented in *N. acidilobi* by five ABC transporter subunit genes and two major facilitator superfamily permeases), detected only at low levels in the proteomic data sets. It is unlikely that the multitude of transporters necessary for metabolite transfer could be packed densely enough at the contact site to provide the flux and broad specificity required to sustain cellular growth and division. *N. acidilobi* and its relatives that inhabit hot springs with pH as low as 2.5 (ref. 25) have an additional adaptation challenge: maintenance of cytoplasmic pH within a biocompatible range, similar to other thermoacidophiles[45]. A candidate for this function in *N. acidilobi* is a membrane P-type ATPase (Nps1260), which we hypothesize could maintain the ΔpH by actively pumping protons out of the cytoplasm thus providing a membrane potential.

The host organism, *Acidilobus* 7A, has a 1.5-Mbp genome, encoding 50 RNAs (rRNAs, tRNAs and RNaseP RNA) and 1,508 proteins. On the basis of mass spectrometry we confirmed the presence and relative abundance of a large fraction of the encoded proteome (71%, or 1,082 proteins; Supplementary Data 3). The genome is nearly identical in size with that of *A. saccharovorans* from Kamchatka[33] and those of several uncultured *Acidilobus* from various thermal springs in YNP[31], with large regions of synteny and common metabolic potential (Fig. 6 and Supplementary Fig. 5). As such, the genome of *Acidilobus* 7A encodes enzymatic and transport functions to hydrolyse and ferment polysaccharides and peptides, which can provide energy by anaerobic oxidation of the resulting organic acids via the tricarboxylic acid cycle. Similar to the other *Acidilobus* species, *Acidilobus* 7A appears to be auxotrophic for a variety of vitamins, purines and amino acids, which would need to be imported. Importantly, none of these auxotrophies are specific to *Acidilobus* 7A as compared with other *Acidilobus* spp., and therefore appear not related to colonization by *N. acidilobi*. The membrane electron transport processes in *Acidilobus* are not well understood, especially the sulfur dependence of *A. sulfurreducens*, as well as the stimulatory effects of sulfur and thiosulfate on the other species including *Acidilobus* 7A. Either a proposed soluble NAD(P)H-dependent sulfur oxidoreductase or a quinone-dependent membrane-bound sulfur-reducing complex have been invoked[32,33], but the biochemical mechanism is questionable[31]. Nevertheless, we detected by proteomics all the annotated membrane bound energy-generating complexes. Collectively, the proteins involved in energy production and conversion account for relatively a quarter of the measured proteome, with several enzymes from the tricarboxylic acid cycle being among the most abundant cellular proteins (Fig. 7 and Supplementary Data 4). Other functional categories present at high relative abundance are chaperones and stress response proteins (peroxiredoxin, thermosome and superoxide dismutase). These are important not only for life at high temperature but also as a response to the metabolic burden imposed by *N. acidilobi*, which could lead to enhanced levels of oxidative stress on the

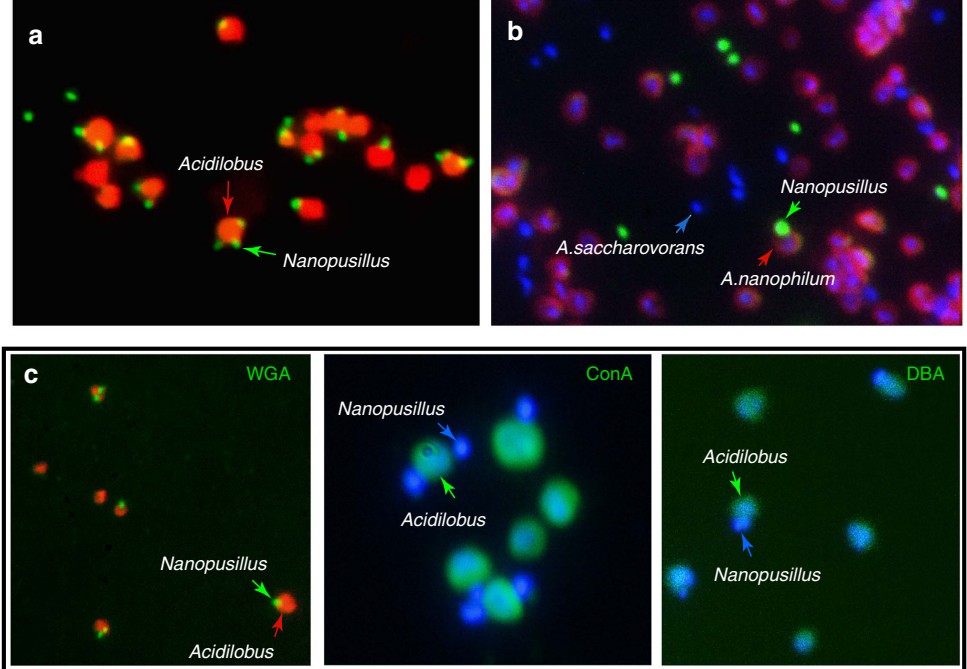

**Figure 5 | Immunofluorescence staining of co-cultures of *N.acidilobi, Acidilobus* sp. 7A and *A. saccharovorans*.** (**a**) *N.acidilobi–Acidilobus* sp. 7A co-culture. (**b**) *N.acidilobi–Acidilobus* sp. 7A–*A. saccharovorans* co-cultures. Blue corresponds to staining of the cell nucleoid DNA by DAPI (the only staining for *A. saccharovorans*). (**c**) Fluorescein-lectin (WGA, ConA and DBA) staining of *N.acidilobi–Acidilobus* sp. 7A co-cultures, counterstained with the DNA-binding dyes Syto62 (red) or DAPI (blue).

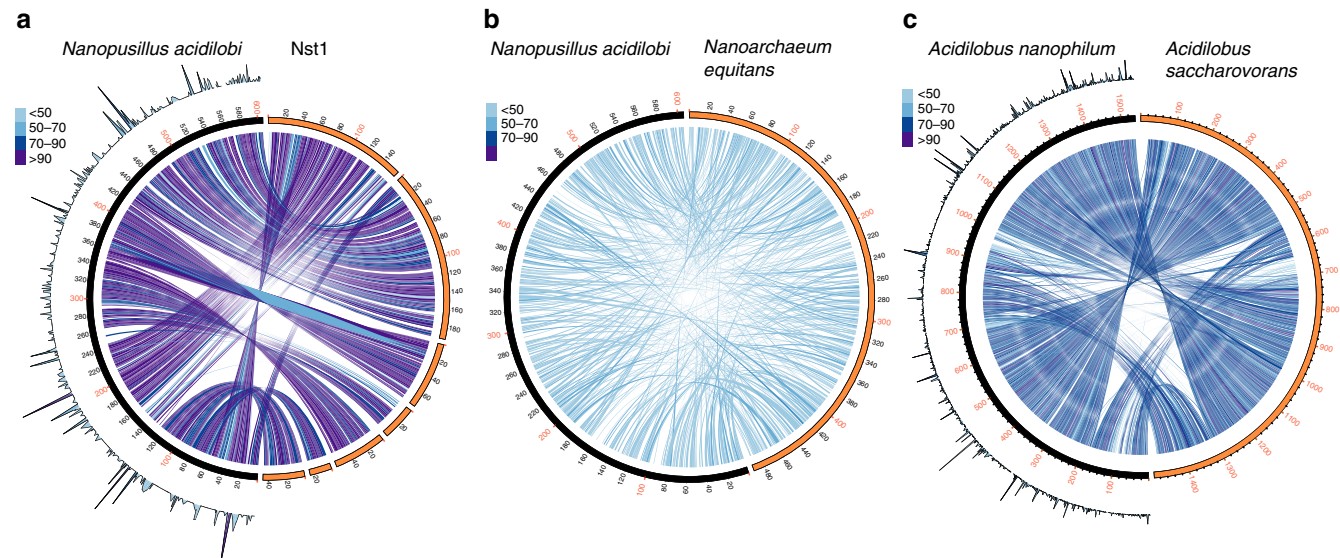

**Figure 6 | Circos-based genome alignments.** *N. acidilobi*-Nst1 (**a**), *N. acidilobi–N. equitans* (**b**) and *Acidilobus* sp. 7A-*A. saccharovorans* (**c**). Connecting line indicate pairs of orthologues between the genomes, the colour being scaled to % amino-acid identity levels. The outer ring histograms (**a**,**c**) show relative abundance of individual encoded proteins based on mass spectrometry proteomics.

system. When comparing the proteomes of the *I. hospitalis–N. equitans* system[18] with that of *Acidilobus* 7A-*N. acidilobi*, host energy-linked enzymes and stress response are dominant in both hosts, even though the two species of Crenarchaeota have very different types of metabolism.

## Discussion

*N. equitans* and *N. acidilobi* illustrate common strategies used by relatively distant members of the Nanoarchaeota for an ectoparasitic lifestyle on hyperthermophilic Crenarchaeota. Unlike bacteria that are intracellular symbionts (mutualists or parasites), Nanoarchaeota have maintained an open environ-ment-exposed cellular membrane and developed a mechanism to acquire all primary biosynthetic molecules from the host cell through a discrete cell–cell contact site. One potential advantage of such strategy might be the option of detachment from a dying host. Archaellum-mediated motility may also enable adaptation to different hosts, shown to occur in Yellowstone among closely related Nanoarchaeota species that use either a Desulfurococcales

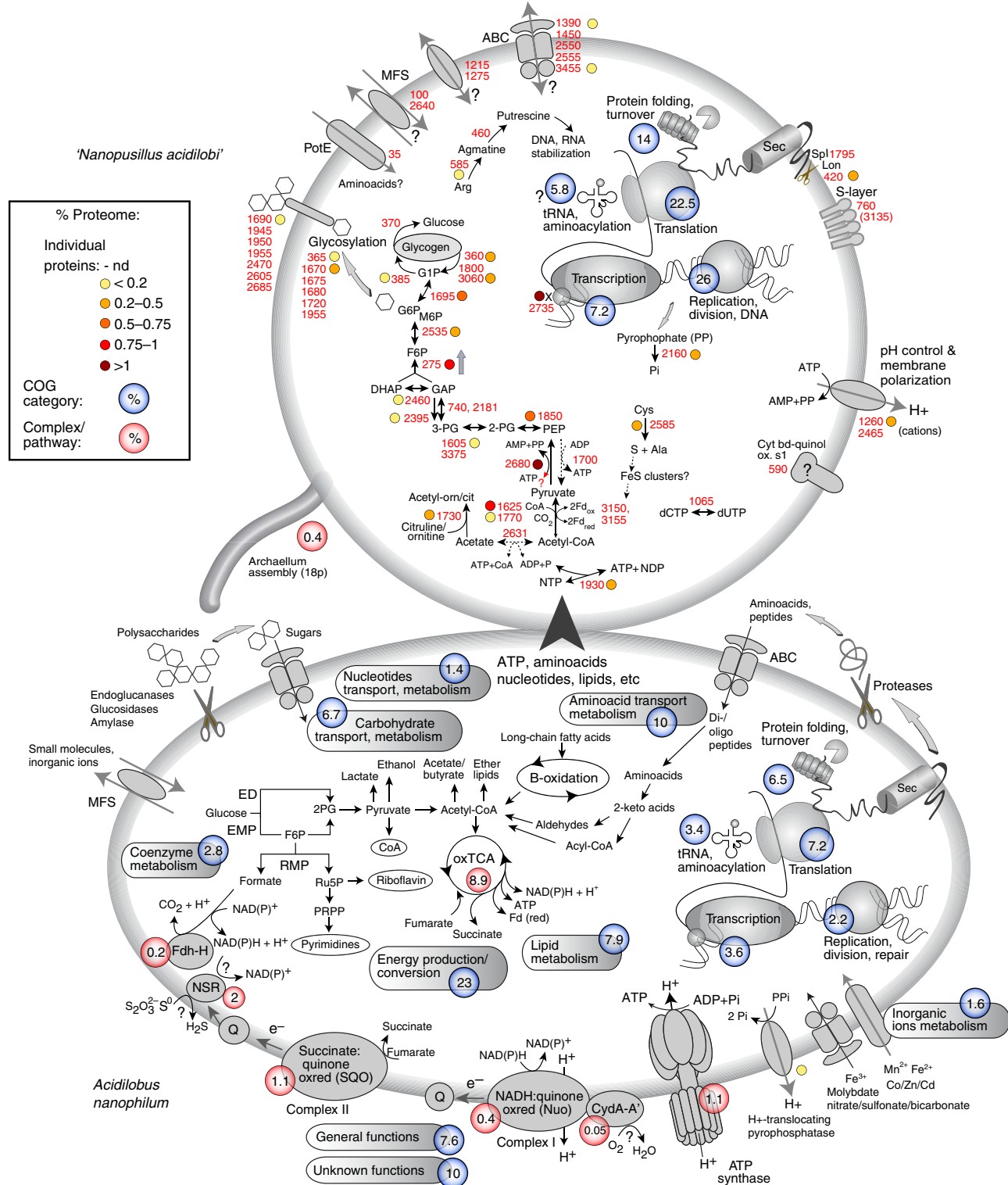

**Figure 7 | Metabolic reconstruction of *N. acidilobi–Acidilobus* sp. 7A based on genomic sequences.** Proteome relative abundance data (based on protein spectral abundance) is colour-scaled for selected individual proteins in *N. acidilobi* or represented combined for functional COG categories (blue spheres) or specific metabolic processes or protein complexes (red spheres), with corresponding relative abundance.

(*Acidilobus*) or a Sulfolobales archaeon. The extremely small size and the strict anaerobic nature of these organisms have precluded, so far, experimental reconstitution of the interaction by inoculation of *N. equitans* and *N. acidilobi* single cells into pure host cultures. Therefore, the viability and 'infectivity' of such detached nanoarchaeal cells, in the lab or the environment, remain unknown.

The genomes of Nanoarchaeota are amongst the smallest for Archaea and Bacteria and, compared with parasitic and symbiotic bacteria, are unique in selectively retaining a full repertoire of genetic information processing systems while eliminating biosynthetic and energetic capabilities. Gene loss occurred possibly through intra-chromosomal recombination and deletion, which may have led to the fragmentation of some protein-encoding

genes that are essential but can function in *trans* by post-translational assembly. The shared complement of split genes, as well as their break points strongly suggests that the process of genome reduction had started before the divergence of the marine and terrestrial nanoarchaeal lineages. Specific association with a nanoarchaeon appears to have impacted the genomes of some of the host organisms through lateral gene transfer and streamlining[22,46]. The host of *N. acidilobi*, however, is relatively unchanged compared with other *Acidilobus*. Considering that *Acidilobus* sp. 7A is not strongly impacted by its nanoarchaeon, unlike *I. hospitalis*, could reflect different levels of adaptive response to such relationships. As no physiologically detectable benefits for the host in either system (at least in laboratory cultures) or genomic-encoded complementation are evident, these 'intimate associations'[19] may range between commensal and ectoparasitic-type symbioses. Characterization of additional nanoarchaeal systems from the global oceanic hydrothermal vent systems and from distant geothermal sites should reveal the diversity of hosts that these organisms can utilize, as well as the range of genomic changes associated with such interspecific associations. In the quest to identify the mechanisms of interspecies interaction it could be important to determine whether the type of host metabolism (for example, autotrophic versus heterotrophic), repertoire of transporters and cell surface characteristics have impacted the degree of genome reduction and remaining physiological potential of their associated Nanoarchaeota. As identifying the host requires its physical recovery with the Nanoarchaeota cell, either single-cell genomics or direct cultivation are essential.

The proteins and/or cellular structures that facilitate the specific interaction between the nanoarchaeal cells and their host, as well as the transfer of metabolites, have remained elusive. Though still enigmatic, it is remarkable that these cells can apparently achieve the transfer, through areas of a few nanometres across, not only of a chemically diverse repertoire of small molecules (lipids, nucleotides, amino acids and ATP) but also at a rate that sustains cell growth and division. Ultrastructural studies combined with molecular characterization of some of the conserved predicted membrane proteins encoded by the genomes of the Nanoarchaeota should eventually unravel that mystery. A similar although evolutionarily independent mechanism may be used by parasitic TM7 bacteria that colonize the surface of human oral *Actinomyces*[47]. As more members of distant bacterial and archaeal taxa are cultivated and characterized, we will develop a better understanding of the diversity of cell–cell interaction mechanisms in the microbial world and how they have evolved across the three domains of life.

## Methods

**Sample collection and culture enrichments.** Sampling was conducted in June 2012 at Cistern Spring pool, in the Norris Geyser basin of YNP (latitude: 44.723; longitude: −110.704"), under the research permit YELL-2008-SCI-5714. Water and sediment slurry samples (82 °C, pH 4.5) were collected in 100-ml sterile bottles and sealed immediately with butyl rubber stoppers displacing excess water through syringe needle so that no air remained on top of the liquid. Samples were stored at 4 °C and the archaeal community composition was characterized using 454 pyrosequencing of the 16S RNA V4 region as described in ref. 48. The archaeal diversity in the Cistern Spring environmental sample, based on 16S rRNA (9,657 sequences), encompassed members of the Nanoarchaeota (7%), Thermofilaceae (3%), Thermoproteaceae (48%), Desulfurococcaceae (2%), Acidilobaceae (11%) and Sulfolobaceae (29%). We hypothesized that the host of some of the Yellowstone Nanoarchaeota would be an organism that could withstand pH values well below those of Obsidian Pool (pH 5.5–6.5) or Cistern Spring (pH 4.5–5), likely an acidophilic archaeaon from Sulfolobaceae or Acidilobaceae. Therefore we aimed at cultivation conditions that would select for such organisms and restrict the growth of neutrophiles.

Archaeal enrichments were set up using basal *Sulfolobus* culture media[26] supplemented with combinations of carbon sources and electron acceptors, containing (per litre): 1.30 g $(NH_4)_2SO_4$; 0.28 g $KH_2PO_4$; 0.25 g $MgSO_4 \cdot 7H_2O$; 70 mg $CaCl_2 \cdot 2H_2O$; 20 mg $FeCl_3 \cdot 6H_2O$; 1.8 mg $MnCl_2 \cdot 4H_2O$; 4.5 mg $Na_2B_4O_7 \cdot 10H_2O$, 0.22 mg $ZnSO_4 \cdot 7H_2O$; 0.05 mg $CuCl_2 \cdot 2H_2O$; 0.03 mg $Na_2M0O_4 \cdot 2H_2O$; 0.03 mg $VOSO_4 \cdot 2H_2O$; and 0.01 mg $CoSO_4$, supplemented with vitamins (DSMZ medium 141), 0.03 g yeast extract and 0.5 g peptone, casamino acids or 0.5 g sucrose. The pH was varied between 2.5 and 4.5. A volume of 1 ml environmental sample was inoculated in 20-ml media, in serum bottles capped with butyl rubber stoppers followed by incubation at 85 °C under non-reduced (without shaking) or reduced conditions (100 µM cysteine), under an atmosphere of 80% $N_2$–20% $CO_2$, for 5–7 days. To extract DNA, 3-ml aliquots were retrieved and filtered through 0.2-µm syringe filters. The cells were washed with phosphate-buffered saline (PBS) and retrieved by back aspiration of a 1-ml cell-lysis solution containing 100 mM TrisCl (pH 7.5), 200 mM NaCl, 5 mM EDTA and 1% SDS. Proteinase K was added to 100 µg ml$^{-1}$ and the lysate incubated for 1 h at 50 °C, followed by addition of SDS to 2% and further incubation for 1 h. The lysate was extracted with phenol-chloroform and the DNA was precipitated overnight at −20 °C following the addition of 50 µl ammonium acetate (5 M), 15 µg GlycoBlue (Ambion) and one volume of isopropanol. The DNA was recovered by centrifugation (10 min at 15,000g), washed with cold 70% ethanol, air dried and dissolved in 20 µl TrisCl 10 mM and EDTA 1 mM (pH 8.0). To monitor the presence and relative abundance of Nanoarchaeota we performed PCR of the 16S rRNA gene with the Nanoarchaeota-specific primers 7mcF-1511mcR (ref. 15), which were confirmed by TOPO-TA cloning (Invitrogen, Carlsbad CA, USA) and Sanger sequencing. The other Archaea present in the enrichments were identified following cloning and sequencing of the 16S rRNA gene using the archaeal universal primers 515AF2-1492R (ref. 48). Fluorescence *in situ* hybridizations were performed on 3% paraformaldehyde-fixed cells, as described in ref. 10, using the 5′AlexaFluor488-labelled Nanoarchaeota probe 515mcR2 (5′-CCCTCTGGCCCCAC TGCT-3′) and 5′AlexaFluor546-labelled Crenarchaeota probe CREN499R (5′-CCA GWCTTGCCCCCCGCT-3′) (Integrated DNA Technologies, Inc., Coralville, USA). Examination by fluorescence microscopy revealed that nanoarchaeal cells were present on the surface of disk-shaped Crenarchaeota in the enrichments.

**Isolation and characterization of *N. acidilobi–Acidilobus* sp.** To identify the host organism for the Nanoarchaeota in enrichments we sorted randomly selected 'single' cells (the nanoarchaeon attached to its host) in two 96-well plates using a Cytopeia Influx Model 208S cell sorter (Cytopeia, Seattle, WA). The genomic DNA was amplified using phi29 DNA polymerase, and the single-amplified genomic (SAG) products were used for amplification of 16S rRNA genes using Nanoarchaeota-specific primers (7mcF-1511mcR). Three positive SAGs were identified and confirmed to represent Nanoarchaeota by Sanger sequencing. PCR of 16S rRNA genes using Archaea primers 515AF2-1492R (which do not recognize Nanoarchaeota) followed by Sanger sequencing established the presence of *Acidilobus* in all Nanoarchaeota-containing SAGs.

Following identification of *Acidilobus*, a comparison of the *Sulfolobus* mineral base medium with the one used for the isolation of *A. sulfurireducens*[28] resulted in an approximately twofold improved growth rate of the *Acidilobus*–Nanoarchaeota population. We therefore switched to using the *Acidilobus* medium, containing (per litre): 0.33 g $NH_4Cl$; 0.33 g $KH_2PO_4$; 0.33 g $MgSO_4 \cdot 7H_2O$; 0.33 g $CaCl_2$; and 0.33 g KCl, supplemented with SL-10 trace metals (1 ml l$^{-1}$), a fivefold excess of Wolfe's vitamins and 0.3 g yeast extract. A variety of carbon sources were tested (0.05% peptone, meat extract, glycogen, sucrose, cellobiose, maltose, trehalose, glucose or fructose), as well as inorganic terminal electron acceptors (elemental sulfur, sodium thiosulfate and nitrate). We also tested the effect of adding volatile fatty acids (1 ml l$^{-1}$ of a solution of isobutyric acid, 2-methylbutyric acid, isovaleric acid and valeric acid, 0.5% each) or filter-sterilized bovine rumen fluid (0.01%). The final pH was adjusted with 2 N $H_2SO_4$. The medium was filter-sterilized and dispensed in sterile vessels followed by degassing using three cycles (20 min each) of vacuum:sterile 80% $N_2$–20% $CO_2$ fill and reduced with 100 µM cysteine at 80 °C overnight. The co-culture could be preserved by freezing at −80 °C in the presence of 10% dimethylsulfoxide.

Subsequent cultivation was performed in the Acidilobus-type medium supplemented with yeast extract (0.03%) and peptone (0.05%) pH 3.6, at 82 °C, under 80% $N_2$–20% $CO_2$. Dilution to extinction resulted in microscopically homogeneous co-cultures of *Acidilobus* sp. 7A and *N. acidilobi*. A final pure co-culture of *Acidilobus* sp. 7A and *N. acidilobi* was obtained using optical tweezer selection with a Zeiss PALM MicroTweezers system (Zeiss, Thornwood, NY). Following growth of the co-culture, an *Acidilobus* sp. 7A cell not harbouring a nanoarchaeon was isolated by optical tweezer and used to establish a pure culture of the host. The effect of temperature and pH was tested using anaerobic *Acidilobus* basal salts medium supplemented with vitamins, 0.03% yeast extract, 0.05% peptone and 0.05% sodium thiosulfate. To determine cell counts we used a qPCR assay (BioRad iQ SYBR Green Supermix) in a CFX96TM cycler (Bio-Rad, Hercules, CA), with 16S rRNA primers specific for *Nanopusillus* (5′-GTGGGCC AGAGGGGTGG-3′ and 5′-TGGCTTCTTCCGTCCCCTAG-3′) or for *Acidilobus* (5′-GGGGCAAGTCTGGTGT-3′ and 5′-GCCTTTCCCGCCCCCTAGC-3′) and with full-length 16S rDNA plasmid clones as standards. Cell density values inferred from qPCR data were validated by direct microscopic counting, using a Petroff Hausser Counting Chamber slide (EMS, Hatfield, PA).

For electron microscopy analyses, *Acidilobus–Nanopusillus* cells were collected by gentle filtration on 0.1-µm filters followed by washing with PBS (pH 7.2) and

fixation with 3% glutaraldehyde in PBS. Subsequent steps were as we described in ref. 18. Gold-coated cells were imaged using a Zeiss Auriga Focused Ion Beam Scanning Electron Microscope.

**Analysis of co-cultures using fluorescent cell labelling.** A custom antiserum against *Acidilobus–Nanopusillus* was raised in rabbits by Covance Inc. (Denver PA), and the IgG was purified by protein G affinity chromatography. We labelled IgG aliquots with AlexaFluor488 or with AlexaFluor647 using the ThermoFisher protein labelling and purification kits. For further fractionation, aliquots of labelled IgG were incubated for 1 h with a suspension of 5–10 mg of *Acidilobus* sp.7A or *A. saccharovorans* cells in 5% goat serum in PBS. Following removal of the cells by centrifugation, the specificity of the remaining IgG was enriched for *Nanopusillus* or *Acidilobus* sp.7A, respectively. For immunofluorescence labelling, fixed cells were incubated on gelatin-coated slides with a blocking solution (5% goat serum in PBS) for 30 min followed by 1-h staining with fluorescent antibody cocktails (diluted 1:20–1:100) in 5% goat serum-PBS and washing with PBS. Slides were mounted in DAPI-containing Citifluor antifade solution (Citifluor Ltd. London) and visualized using a Zeiss AxioImager fluorescence microscope.

For studying the host specificity of *Nanopusillus*, actively growing *Acidilobus* sp.7A–*N. acidilobi* co-cultures were mixed at various ratios with cultures of *A. saccharovorans* DSM 16705 (ref. 29) and incubated for 2 days at 80 °C. Cells were collected by filtration on 0.1-mm filters, fixed and analysed by immunofluorescence labelling with labelled antibodies (*Nanopusillus*-enriched AlexaFluor488 IgG and *A. saccharovorans*-depleted AlexaFluor647 *Acidilobus* IgG) and with the DNA-binding dye DAPI.

For identifying the presence of cell surface glycoproteins and/or glycolipds, fixed *Acidilobus* sp.7A–*N. acidilobi* cells were immobilized on poly-lysine-coated slides and stained with individual, fluorescein-conjugated lectins ConA, SBA, WGA, DBA, UEA-I, RCA or PNA ($20\,\mu g\,ml^{-1}$) in PBS (Kit1, Vector Laboratories Inc., Burlingame, CA) for 30 min followed by PBS washing and mounting in antifade solution containing DNA-binding dyes (DAPI or Syto62). Live cells were also analysed by staining and visualization on top of 0.2-μm black polycarbonate filters, with similar results.

**Genomic sequencing, annotation and analysis.** Genomic DNAs were used to generate Nextera DNA libraries followed by sequencing on an Illumina MiSeq instrument, using $2 \times 300$-nucleotide paired-end chemistry kit, according to the manufacturer's instructions. Preliminary assembly and composition-based binning was performed as in ref. 22. To obtain sufficient *N. acidilobi* genomic DNA for PacBio sequencing we purified it from the host DNA using ultracentrifugation in a self-forming gradient of CsCl in the presence of the $A + T$ binding dye bisbenzimide (Hoechst 33258) as described in ref. 49. Approximately 10 μg purified genomic DNA was used for generating 8–10 kb insert libraries followed by single-molecule, real-time sequencing on a PacBio RS II instrument at the Genomics Resource Center of the University of Maryland Institute for Genome Science. Hybrid assemblies of the PacBio and MiSeq data were performed as described[50]. The closed genomes of *Acidilobus* sp 7A and *N. acidilobi* were annotated by the NCBI Prokaryotic Genome Annotation Pipeline (2013 release). Phylogenetic analyses, metabolic reconstructions and comparative genomics with other Nanoarchaeota and publicly available *Acidilobus* genomic and metagenomic data sets[21,22,25,31,33,46] were performed based on functional annotations and archaeal COG classification[51], as in ref. 22. Genomic alignments were displayed using Circos[52].

**Proteomic analysis.** Three cell pellets (~10 mg each) of mid-log co-cultures of *Acidilobus* sp 7A–*N. acidilobi* were combined and resuspended in lysis buffer (4% SDS, 100 mM Tris-HCl (pH 8.0) and 50 mM dithiothreitol) and incubated for 10 min at 100 °C followed by sonic disruption (Branson Sonifier). Extracted proteins were then precipitated with 20% trichloroacetic acid on ice, pelleted, washed with cold acetone and air-dried. Proteins were re-solubilized in 8 M urea, 5 mM dithiothreitol, 100 mM Tris-HCl, pH 8.0 and digested with trypsin as described in ref. 17. Tryptic peptides were analysed using MudPIT LC-MS/MS over multiple salt-cuts (with subsequent 2 h reversed-phase separation per cut) using a hybrid LTQ Orbitrap Pro (ThermoFisher) mass spectrometer operating in data-dependent acquisition as previously described[17,18]. Resulting MS/MS spectra from four technical replicate runs were matched to computationally predicted tryptic peptides gleaned from the supplied proteome FASTA database (*Acidilobus* sp.7A and *N. acidilobi* protein sequences appended with common contaminants and decoy sequences to assess false-discovery rates) using MyriMatch v. 2.1 (ref. 53). Peptide-spectral matches were filtered (peptide- and protein-level false-discovery rates tested down to <1%) and assembled to their respective proteins using IDPicker v. 3.0 (ref. 54). Both normalized spectral counts[17,18] and ion intensities were used for relative quantification. Peptide ion intensity values were derived from either MS1/parent ion intensity (area under the curve) via IDPicker or matched-ion intensity[55]. Resulting peptide intensity distributions were log2-transformed, normalized across replicates (LOESS), and standardized by median absolute deviation and median centring using InfernoRDN[56]. Protein abundances were derived using the RRollup method as previously described[55]. Supplementary Data 1–4 list the predicted proteins of both organisms and provide a comparison of

relative abundance data for individual proteins, with rank levels between the different analyses.

**Data availability.** Genomic sequences are available in GenBank under the accession numbers CP010514 and CP010515. Proteomic data were deposited at the MassIVE data repository (accession numbers: MSV000079705 (MassIVE) and PXD004102 (ProteomeXchange)). The authors declare that all other data supporting the findings of this study are available within the article and its Supplementary Information files, or from the corresponding author on request.

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

## Acknowledgements

This research was supported by grants from the National Science Foundation (DEB1134877) and from the U.S. Department of Energy, Office of Biological and Environmental Research (DE-SC0006654). Oak Ridge National Laboratory is managed by UT-Battelle, LLC, for the U.S. Department of Energy under contract DE-AC05-00OR22725. We thank the administration of Yellowstone National Park for the permit YELL-2008-SCI-5714 and Christie Hendrix and Stacey Gunther for coordinating the sampling activities. We acknowledge the University of Tennessee Advanced Microscopy and Imaging Center for instrument use, scientific and technical assistance with scanning electron microscopy. We acknowledge the Genomics Resource Center of the University of Maryland Institute for Genome Science for PacBio sequencing and the Genomics Core at the University of Tennessee Knoxville for Sanger sequencing. We also thank Steve Allman, Zamin Yang and Dawn Klingeman for assistance with cell sorting, molecular biology techniques and MiSeq sequencing.

## Author contributions

L.W.: lab experiments (cultivations, DNA extraction, qPCR, fluorescence microscopy and scanning electron microscopy); R.J.G.: lab experiments (proteomic sample preparation and mass spectrometry data acquisition), bioinformatics (proteomic data analysis) and wrote the manuscript; B.S.B.: lab experiments (cultivations, DNA extractions and fluorescence microscopy); C.S.: lab experiments (sampling, cultivation, DNA extractions and qPCR); S.U.: bioinformatics (genome assembly and annotation, and comparative genomics); R.L.H.: bioinformatics (proteome analysis); A.-L.R.: wrote the manuscript. M.P.: lab experiments (sampling, cultivations, DNA isolation and fluorescence microscopy), bioinformatics (genome annotation, metabolic pathway analysis and comparative genomics) and wrote the manuscript.

## Additional information

**Competing financial interests**: The authors declare no competing financial interests.

**How to cite this article**: Wurch, L. *et al.* Genomics-informed isolation and characterization of a symbiotic Nanoarchaeota system from a terrestrial geothermal environment. *Nat. Commun.* 7:12115 doi: 10.1038/ncomms12115 (2016).

