## [Transparent Peer Review · Nature Communications]

Reviewers' comments:

Reviewer #1 (Remarks to the Author):

In the manuscript "From gnomes to cultures: a parasitic Nanoarchaeota system from a Yellowstone geothermal spring" by Wurch et al., the authors report the identification of a new member of the Nanoarchaeota "Nanopusillus acidilobi" and its host "Acidilobus sp. 7A. This is an important novel finding that highlights the microbial diversity and provides some new insights into the evolutionary history of Nanoarchaeota, which are characterized by their unusual lifestyle that is still not fully understood.

The authors were able to isolate N. acidilobi and its host by the combination of cultivation experiments and single cell genomics and succeeded to retrieve complete genome information as well as proteomics data. In addition to Nanoarchaeum equitans and Nst1 described previously (some of the authors are also involved in this study), this is only the third member of the Nanoarchaeota that can be cultivated and valuable insight into the genomic and physiological complexity of the organism are presented.

Therefore, this manuscript provides new exciting insights into the microbial diversity with important phylogenetic implications and thus represents a very important contribution in the archaeal/microbial diversity field (top five this year). From the experimental point of view this is an almost complete piece of work that comprises an elaborate microbial, genetic and proteomic analyses of two organisms. The conclusion drawn are besides some minor points appropriate. The manuscript is clearly written, only in some cases I would propose changes for clarity (e.g. parasitic/symbiotic life style, see below for discussion). The citation of references is appropriate only at some positions, the authors may want to consider to include some citations for previously characterized enzyme homologs (see below for details).

P2, L7: "ectoparasitic host" As far as I understand it is still a matter of debate if Nanoarchaeota are parasites or symbionts. Also in the manuscript Nanopusillus acidilobi is referred to as ectoparasite, whereas Nanoarchaeum equitans is called ectosymbiont (P3,L2) with parasitic adaptation (P3, L8; also P3, L25, 28 novel symbiotic system, mechanism of symbiosis etc.) For the

reader it would be helpful to discuss/clarify this point and to provide additional information about the reasons for this different classification.

P5, L14 ff; P6, L23: "Occasional free cells can be observed in the co-culture but their viability is unknown." In combination with the presence of the archaellum as motility structure this is a very interesting point. Did the author's check the viability by isolation of the free *Nanopusillus acidilobi* cells and incubation with a pure *Acicilobus* sp. 7A culture? Are the cells able to attach and grow? This might be at least a first evidence for viability.

P8, L8ff: The absence of the classical enzyme couple PGK/GAPDH, key enzymes of gluconeogenesis in (hyper)thermophilic Archaea, is quite interesting. However, the GAPN is described as unidirectional, non-phosphorylating enzyme and therefore is regarded to be involved in glycolysis only. In many anaerobic hyperthermophiles in addition to GAPN a reversible ferredoxin-dependent GAP oxidoreductase is found [1] and more recently new glyceraldehyde ORs were identified in *S. tokodaii* [2].

In respect to the missing pyruvate kinase, the PEPS from *T. kodakarensis* possesses glycolytic activity with a major function in glycolysis [3]. In addition pyruvate,phosphate dikinase (PPDK) catalyses both reactions and the reversible enzyme shows high similarity to PEPs. However, there have been some residues determined that allow to differentiate between both enzymes [4]. The author's may want to reanalyze their sequences considering this information (for recent review on carbohydrate metabolism in Archaea see [5]).

Regarding the formation of glycogen as carbon storage it would of course make sense to possess the ability for degradation, therefore it is an interesting question if the glycolytic pathway is active. May be the pathway is only partially active and glycogen is used as source for pentose formation (rather than forming pyruvate). Is the reversed ribulose monophosphate pathway present?

P8, L27 There are different TrmBLs described and besides the one involved in sugar regulation one homolog TrmBL2 in *Tko* is reported as abundant chromosomal protein and global transcriptional repressor [6].

P10,L4: Regarding the membrane proteins in respect to transport and exchange of metabolites it would be interesting to comment on the transporters found (e.g. number of ABC transporters) to get an idea what "minute repertoire" means.

P10 L10: For the discussion of the additional challenge exposed by the low pH environments compared to *N. equitans* it would also make sense to consider the lifestyle of the host (heterotrophic/autotrophic, respectively).

Supplementary Table 2b: The "% proteome spectra" should be given like in Table 2d.

Literature:

1. Mukund, S. & Adams, M. W. W. (1995) Glyceraldehyde-3-phosphate ferredoxin oxidoreductase, a novel tungsten- containing enzyme with a potential glycolytic role in the hyperthermophilic archaeon *Pyrococcus furiosus*, *Journal of Biological Chemistry*. 270, 8389-8392.
2. Wakagi, T., Nishimasu, H., Miyake, M. & Fushinobu, S. (2016) Archaeal Mo-Containing Glyceraldehyde Oxidoreductase Isozymes Exhibit Diverse Substrate Specificities through Unique Subunit Assemblies, *PloS one*. 11, e0147333.
3. Imanaka, H., Yamatsu, A., Fukui, T., Atomi, H. & Imanaka, T. (2006) Phosphoenolpyruvate synthase plays an essential role for glycolysis in the modified Embden-Meyerhof pathway in *Thermococcus kodakarensis*, *Molecular microbiology*. 61, 898-909.
4. Tjaden, B., Plagens, A., Dorr, C., Siebers, B. & Hensel, R. (2006) Phosphoenolpyruvate synthetase and pyruvate, phosphate dikinase of *Thermoproteus tenax*: key pieces in the puzzle of archaeal carbohydrate metabolism, *Molecular microbiology*. 60, 287-98.
5. Bräsen, C., Esser, D., Rauch, B. & Siebers, B. (2014) Carbohydrate Metabolism in Archaea: Current Insights into Unusual Enzymes and Pathways and Their Regulation, *Microbiology and Molecular Biology Reviews*. 78, 89-175.

6. Maruyama, H., Shin, M., Oda, T., Matsumi, R., Ohniwa, R. L., Itoh, T., Shirahige, K., Imanaka, T., Atomi, H., Yoshimura, S. H. & Takeyasu, K. (2011) Histone and TK0471/TrmBL2 form a novel heterogeneous genome architecture in the hyperthermophilic archaeon *Thermococcus kodakarensis*, *Molecular biology of the cell*. 22, 386-98.

Reviewer #2 (Remarks to the Author):

Summary of the key results

This manuscript describes a new ectoparasitic Nanoarchaeota, including some of its cell biology, genome sequence and proteome. The authors look for an explanation for how such organisms might be able to obtain relevant nutrients from their hosts (they lack the machinery to synthesize their own so presumably transport it) but do not find definitive evidence for a mechanism.

Originality and interest

The biology around ectoparasitic cells, especially amazingly reduced cells such as these, is fascinating!

Data & methodology

My expertise is in proteomics so I will focus on that. Just generally though, the other methods seem to be appropriately chosen and applied. For the proteomics, the raw data absolutely must be made available at ProteomeXchange before this can be considered for publication. I could find no evidence that the authors had submitted these data anywhere. The FDR they use (5%) is no longer used in the proteomics field so they should apply a 1% FDR and then re-analyze their data in that light.

Similarly, the % Proteome Spectra values the authors use aren't useful in any way. At the VERY least, such values need to be normalized to the number of possible observable tryptic peptides for each protein. Far better than this should would be an ion intensity-based estimate of relative abundance. The authors use a somewhat arcane software package for MS/MS identification so I'm not familiar with its capabilities. Something like MaxQuant would work

with these data though and it is open access and produces ion intensity values.

Appropriate use of statistics and treatment of uncertainties

This could be mentioned here or in the above section: there does not appear to be any replication in the proteomics data so that values presented (even if ion intensity measures are extracted) are of questionable data.

Conclusions: robustness, validity, reliability

The findings of common evolutionary history and adaptations to *Nanoarchaeum equitans* may be true but are not robustly supported by the proteomics yet. The above suggestions should help to make this much more solid.

Reviewer #3 (Remarks to the Author):

In this manuscript the authors describe the identification and cultivation of a new member from the Nanoarchaeota group. A non marine one.

This is simply an amazing manuscript. I have but one minor comment. Please add to Fig.6 a Circos-based genome alignments between *Nanopusillus acidilobii* and *Nanoarchaeum equitans*.

Responses to reviewers' remarks

We would like to thank all reviewers for their enthusiasm and for their detailed and constructive criticism and suggestions, which were valuable for improving the quality and clarity of the manuscript. Following are individual responses to each raised point. For changes in the manuscript text we used a red font to make it easier to follow.

Reviewer #1 (Remarks to the Author):

In the manuscript "From genomes to cultures: a parasitic Nanoarchaeota system from a Yellowstone geothermal spring" by Wurch et al., the authors report the identification of a new member of the Nanoarchaeota "Nanopusillus acidilobi" and its host "Acidilobus sp. 7A. This is an important novel finding that highlights the microbial diversity and provides some new insights into the evolutionary history of Nanoarchaeota, which are characterized by their unusual lifestyle that is still not fully understood.

The authors were able to isolate *N. acidilobi* and its host by the combination of cultivation experiments and single cell genomics and succeeded to retrieve complete genome information as well as proteomics data. In addition to *Nanoarchaeum equitans* and *Nst1* described previously (some of the authors are also involved in this study), this is only the third member of the Nanoarchaeota that can be cultivated and valuable insight into the genomic and physiological complexity of the organism are presented.

Author comment: We thank the reviewer for the appreciation of the novelty and impact of this work. Actually, Nst1 was only characterized by single cell genomics, therefore N. acidilobi is the second Nanoarchaeota in culture (and the first non-marine one). We indicated that Nst1 is not yet cultured in the revised manuscript, page 3.

Therefore, this manuscript provides new exciting insights into the microbial diversity with important phylogenetic implications and thus represents a very important contribution in the archaeal/microbial diversity field (top five this year). From the experimental point of view this is an almost complete piece of work that comprises an elaborate microbial, genetic and proteomic analyses of two organisms. The conclusion drawn are besides some minor points appropriate. The manuscript is clearly written, only in some cases I would propose changes for clarity (e.g. parasitic/symbiotic life style, see below for discussion). The citation of references is appropriate only at some positions, the authors may want to consider to include some citations for previously characterized enzyme homologs (see below for details).

Author response: Thank you for your appreciation and advice on improving the analyses, which we have addressed below. We also included additional suggested citations.

P2, L7: "ectoparasitic host" As far as I understand it is still a matter of debate if Nanoarchaeota are parasites or symbionts. Also in the manuscript Nanopusillus acidilobi is referred to as ectoparasite, whereas Nanoarchaeum equitans is called ectosymbiont (P3,L2) with parasitic adaptation (P3, L8; also P3, L25, 28 novel symbiotic system, mechanism of symbiosis etc.) For the reader it would be helpful to discuss/clarify this point and to provide additional information about the reasons for this different classification.

Author response: Categorizing the type of relationship between the Nanoarchaeum and Ignicoccus (and now Nanopusilus-Acidilobus) has been difficult, and thus Jahn et al (2008) termed it an "intimate association", a rather neutral term. There is no evidence that the host organism benefits from the association with the nanoarchaeon. At the same time, at least in laboratory cultures, data shows some inhibitory effects on the host, particularly at high nanoarchaeon-host ratios. Metabolic, genomic and functional genomic data also show that the nanoarchaeon absolutely relies on its host for cellular

precursors and energetic molecules. Symbiosis as a general term encompasses all types of interspecies relationships (including parasitism), although it is sometimes assumed to refer to mutualism. Until any evidence of the host benefiting of the association, the nanoarchaea appear to be at best ecto-commensals and more likely ecto-parasites (both types of ectosymbiosis). We made changes in the manuscript to make this more clear and consistent (page 12).

P5, L14 ff; P6, L23: "Occasional free cells can be observed in the co-culture but their viability is unknown." In combination with the presence of the archaellum as motility structure this is a very interesting point. Did the author's check the viability by isolation of the free *Nanopusillus acidilobi* cells and incubation with a pure *Acidilobus* sp. 7A culture? Are the cells able to attach and grow? This might be at least a first evidence for viability.

*Author response: The reconstitution of the system by transfer of a free nanoarchaeon into a host culture has been one of the most important but yet unanswered question in Nanoarchaeota biology for over a decade. Such reconstitution has not been achieved despite multiple efforts with *Nanoarchaeum equitans* (Harald Huber and Karl Stetter, personal communications). Even though free individual cells of *N. equitans* can be manipulated using optical tweezers, with very low throughput, they have not so far yielded a reconstituted system in *Ignicoccus* cultures. The problem is these organisms are strict anaerobes and it is possible they are inactivated by the handling and laser exposure, so those experiments have not been very informative. In the case of *Nanopusillus*, the individual free cells are below the optical resolution in the optical tweezer microscope, even under dark phase. We only observe them by fluorescence in fixed, immobilized specimens, so unfortunately the experiment that the reviewer proposes is not technically feasible at this time. We have updated the text on this aspect (page 12).*

P8, L8 ff: The absence of the classical enzyme couple PGK/GAPDH, key enzymes

of gluconeogenesis in (hyper)thermophilic Archaea, is quite interesting. However, the GAPN is described as unidirectional, non-phosphorylating enzyme and therefore is regarded to be involved in glycolysis only. In many anaerobic hyperthermophiles in addition to GAPN a reversible ferredoxin-dependent GAP oxidoreductase is found [1] and more recently new glyceraldehyde ORs were identified in *S. tokodaii* [2].

In respect to the missing pyruvate kinase, the PEPS from *T. kodakarensis* possesses glycolytic activity with a major function in glycolysis [3]. In addition pyruvate, phosphate dikinase (PPDK) catalyses both reactions and the reversible enzyme shows high similarity to PEPs. However, there have been some residues determined that allow to differentiate between both enzymes [4]. The author's may want to reanalyze their sequences considering this information (for recent review on carbohydrate metabolism in Archaea see [5]). Regarding the formation of glycogen as carbon storage it would of course make sense to possess the ability for degradation, therefore it is an interesting question if the glycolytic pathway is active. May be the pathway is only partially active and glycogen is used as source for pentose formation (rather than forming pyruvate). Is the reversed ribulose monophosphate pathway present?

Author response: Thank you, this information has been quite helpful and we revisited the sequence based on these articles. The Nanopusillus genome indeed encodes a protein (Nps740) related to the reversible ferredoxin-dependent GAP oxidoreductases from Euryarchaeaota and Crenarchaeaota and one that appears a distant relative (Nps2181). We did not find homologues of the S. tokodaii GAOR genes. The only PEPS family enzyme encoded in the genome is distinctively PEPS and not PPDK, with residues characteristic of both Crenarchaeaota (Thermoproteales and Sulfolobales) and Euryarchaeaota (Thermococcales) enzymes. As the Thermococcus kodakarensis PEPS has been shown to be functioning in glycolysis, it may have such role in N. acidilobus as well. The ribulose monophosphate pathway is absent in Nanopusillus. As we discussed, it is also possible that the major role for synthesizing and storing sugars could be to provide the substrates for protein and lipid glycosylation. We modified the manuscript text (page 8) and Figure 7 to reflect these findings and

incorporated those references.

P8, L27 There are different TrmBLs described and besides the one involved in sugar regulation one homolog TrmBL2 in Tko is reported as abundant chromosomal protein and global transcriptional repressor [6].

Author response: We analyzed the Nanopusillus gene (Nps2735) in comparison with the different types of archaeal TrmBL and found that it is most similar to Tko and Pfu TrmBL2 sequences (Tk0471 and PF0496) and therefore involved in chromosome architecture and transcription regulation. We modified the text (page 9) and figure accordingly. Thank you.

P10,L4: Regarding the membrane proteins in respect to transport and exchange of metabolites it would be interesting to comment on the transporters found (e.g. number of ABC transporters) to get an idea what "minute repertoire" means.

Author response: We provided more detail on that in the manuscript (page 10) and figure 7, as requested.

P10 L10: For the discussion of the additional challenge exposed by the low pH environments compared to N equitans it would also make sense to consider the lifestyle of the host (heterotrophic/autotrophic, respectively).

Author response: Indeed. We touched on that in the Discussion, page 13.

Supplementary Table 2b: The "% proteome spectra" should be given like in

Table 2d.

Author response: The proteome data tables have been updated, also as a result of the reviewer #2's comments and requests.

Reviewer #2 (Remarks to the Author):

Summary of the key results

This manuscript describes a new ectoparasitic Nanoarchaeota, including some of its cell biology, genome sequence and proteome. The authors look for an explanation for how such organisms might be able to obtain relevant nutrients from their hosts (they lack the machinery to synthesize their own so presumably transport it) but do not find definitive evidence for a mechanism.

Originality and interest

The biology around ectoparasitic cells, especially amazingly reduced cells such as these, is fascinating!

Author comment: Thank you!

Data & methodology

My expertise is in proteomics so I will focus on that. Just generally though, the other methods seem to be appropriately chosen and applied. For the proteomics, the raw data absolutely must be made available at ProteomeXchange before this can be considered for publication. I could find no evidence that the authors had submitted these data anywhere.

Author response: We have submitted the raw data to MassIVE (accession: MSV000079705) and ProteomeXchange (accession: PXD004102) and available via this link: <ftp://MSV000079705@massive.ucsd.edu> (temporarily the password 'a' may be required for the FTP distribution portal). The text in the M&M has been updated to incorporate this information (page 18).

The FDR they use (5%) is no longer used in the proteomics field so they should apply a 1% FDR and then re-analyze their data in that light.

Author response: From our experience working with non-standard genome sizes, i.e. both very small (such as this study) and very large (metaproteomes), the FDR calculation at the protein level does not sufficiently capture false-discovery. Thus, we pay more attention to peptide-level FDR as we feel this better represents what is actually being measured and is not an artifact of protein inference. That said, we have adjusted FDR values of both peptide- and protein-level identifications to < 1% as requested by the reviewer. As most false-hits surface towards the 'bottom' of the data set (i.e. proteins of very low abundance), the abundant proteins and biological processes discussed in the paper were not impacted at all. Both the text and supplementary tables have been updated accordingly to also indicate the new FDR filters of 1%.

Similarly, the % Proteome Spectra values the authors use aren't useful in any way. At the VERY least, such values need to be normalized to the number of possible observable tryptic peptides for each protein. Far better than this should would be an ion intensity-based estimate of relative abundance. The authors use a somewhat arcane software package for MS/MS identification so I'm not familiar with its capabilities. Something like MaxQuant would work with these data though and it is open access and produces ion intensity values.

Appropriate use of statistics and treatment of uncertainties

This could be mentioned here or in the above section: there does not appear

to be any replication in the proteomics data so that values presented (even if ion intensity measures are extracted) are of questionable data.

Author response: We respectfully disagree with the reviewer calling the data 'questionable', especially as the proteomics used in this paper was more so a qualitative analysis and did not delve into biological comparisons across conditions or time points (which would necessitate ample replication indeed). Even with a single run, with appropriate filters and < 10 ppm mass accuracy, the resulting protein list is far from questionable at the top end of protein abundance, which was the main proteomic discussion focus in this study. However, we understand the reviewers concerns and to address that we ran three additional replicates (n=4 total). Furthermore, we supplemented the analysis with normalized spectral counts (NSAF-based which takes protein length into account) and included two forms of ion intensity measurements, both MS1-based area-under-the-curve and matched-ion intensity, as requested. Peptide intensity distributions were normalized via InfernoRDN and protein inference performed with these new ion intensity data. Abundance data from both normalized spectral counts and ion intensities were then compared and presented alongside one another in Supplemental Table 2, with relative standard deviation values based on replicates. The text in the M&M was updated accordingly (Pages 17-18). Overall, the most abundant identified proteins are shared between the various quantitative analyses, even though, as expected they are not precisely the same. Very tight standard deviation between replicates supports robust relative quantification and the differences observed between the different methods are likely linked to the various ways of computing abundance (discrete PSMs vs. MS1 peptide area under the curve vs. matched fragment ion intensities). We thought it appropriate to provide all three so that readers could evaluate the data under multiple quantitative metrics.

With regard to the statement on the use of an 'arcane' software package, there are numerous software packages and pipelines for proteomic data interrogation, as evident in the proteomics literature. We have extensive

experience using and publishing with MyriMatch and IDPicker (developed by Dave Tabb), and thus have designed tools around that platform. Thus, we respectfully disagree that this is an “arcane software package for MS/MS identification” as a literature search reveals over 300 published citations (Google Scholar), many outside our group, using these tools. While MaxQuant is certainly valuable, there are several other comparable tools that provide ion abundance / intensity values.

Conclusions: robustness, validity, reliability

The findings of common evolutionary history and adaptations to Nanoarchaeum equitans may be true but are not robustly supported by the proteomics yet. The above suggestions should help to make this much more solid.

Author response: The reviewer raises some general concerns about the approach and significance of the proteomic analysis. As explained above, we performed replicate analyses and alternative raw data treatments that address most of those concerns. We would like however to also point out some of the goals and rationales. First, the proteomic analysis was not intended to be quantitative and thus we have used relative abundance terms throughout. There are experimental limitations imposed by the microbial system at this point, including the very low biomass available, and the difficulty of obtaining synchronized biological replicates for these organisms, which are still rather “wild”. This proteomic measurement of the new nanoarchaeal system was primarily intended to give an initial glimpse into the main biological processes that are predominant and active. We focused not on relative abundance values for individual proteins, but how they correlate and reveal which biological processes are active and have a relative preponderance in the cellular metabolism. If the majority of the enzymes involved in a particular process are present at relatively high level, regardless of their individual values, we would argue that they indicate that particular process is taking place and that provides the foundation for more focused follow up studies. Also, the strong correlation between the Nanopusillus and Nanoarchaeum proteomic data

supports the similar relative abundance of many of their proteins. Because the Nanoarchaeum proteomic data was analyzed and published back in 20011 using spectral counts (before we switched to ion intensities), we made the comparison using that type of data but have provided all values to address reviewer's concerns. We certainly intend to follow up this study with more focused quantitative functional genomic investigations of these organisms.

Reviewer #3 (Remarks to the Author):

In this manuscript the authors describe the identification and cultivation of a new member from the Nanoarchaeota group. A non marine one.

This is simply an amazing manuscript. I have but one minor comment. Please add to Fig.6 a Circos-based genome alignments between Nanopusillus acidilobii and Nanoarchaeum equitans.

Author response: Thank you! We have incorporated a Circos alignment with Nanoarchaeum equitans in the revised Figure 6 and provided additional discussion in the manuscript on the relatively low level of sequence similarity and sytheny between the two genomes.

REVIEWERS' COMMENTS:

Reviewer #1 (Remarks to the Author):

I'M very happy with the changes and have no more comments regarding the manuscript.

Reviewer #2 (Remarks to the Author):

The authors have adequately addressed my concerns.